# Dynamics of Remote Communication: Movement Coordination in Video-Mediated and Face-to-Face Conversations

**DOI:** 10.3390/e24040559

**Published:** 2022-04-15

**Authors:** Julian Zubek, Ewa Nagórska, Joanna Komorowska-Mach, Katarzyna Skowrońska, Konrad Zieliński, Joanna Rączaszek-Leonardi

**Affiliations:** 1Human Interactivity and Language Lab, Faculty of Psychology, University of Warsaw, 00-927 Warsaw, Poland; ewa.nagorska@psych.uw.edu.pl (E.N.); j.komorowska-mach@uw.edu.pl (J.K.-M.); katarzyna.skowronska@student.uw.edu.pl (K.S.); konrad.zielinski@psych.uw.edu.pl (K.Z.); raczasze@psych.uw.edu.pl (J.R.-L.); 2Faculty of Philosophy, University of Warsaw, 00-927 Warsaw, Poland

**Keywords:** remote communication, movement coordination, recurrence quantification analysis

## Abstract

The present pandemic forced our daily interactions to move into the virtual world. People had to adapt to new communication media that afford different ways of interaction. Remote communication decreases the availability and salience of some cues but also may enable and highlight others. Importantly, basic movement dynamics, which are crucial for any interaction as they are responsible for the informational and affective coupling, are affected. It is therefore essential to discover exactly how these dynamics change. In this exploratory study of six interacting dyads we use traditional variability measures and cross recurrence quantification analysis to compare the movement coordination dynamics in quasi-natural dialogues in four situations: (1) remote video-mediated conversations with a self-view mirror image present, (2) remote video-mediated conversations without a self-view, (3) face-to-face conversations with a self-view, and (4) face-to-face conversations without a self-view. We discovered that in remote interactions movements pertaining to communicative gestures were exaggerated, while the stability of interpersonal coordination was greatly decreased. The presence of the self-view image made the gestures less exaggerated, but did not affect the coordination. The dynamical analyses are helpful in understanding the interaction processes and may be useful in explaining phenomena connected with video-mediated communication, such as “Zoom fatigue”.

## 1. Introduction

When two people engage in a dialogue, they do much more than just exchanging strings of words. According to Fusaroli et al. [1], dialogue participants coordinate on multiple levels, establishing a functional organization fit to a particular situation. Essentially, they form a coupled system within which meanings are co-created, and interaction dynamics are essential to this process [2]. The ability to coordinate movements during interaction is already present in infancy [3] and constitutes the most basic form of bonding with others [4]. Movement coordination allows the establishment of informational and affective coupling [5,6]. This has consequences for various processes of social cognition. As demonstrated by numerous empirical studies, spontaneous movement coordination of people engaged in natural conversations can predict rapport [7], affiliation [8], empathic accuracy [9], joint-action task performance [10,11] or psychotherapy outcomes [12]. The connections between movement coordination and social interaction may go in both directions: particular patterns of movement coordination may be constitutive factors for the interaction or they can be merely indicators of a successful interaction taking place [13]. In any case, by analyzing interpersonal movement coordination, we can infer much regarding the quality of an interaction.

In the present pandemic, many social interactions have moved online. Remote video calls are used as an alternative to face-to-face conversations, both in professional and casual contexts. Video-mediated interactions indeed allow the use of visual cues (gestures, face expressions, body posture) and the establishment of some form of functional movement coordination between participants, which is not possible in audio-only interactions. Studies comparing video-mediated communication to audio-only communication report benefits such as increased effectiveness of group problem-solving, shorter discussion time, and increased emotional bonding [14,15]. However, the experience with video-mediated interactions is not always smooth. In some cases, people were more satisfied with audio-only interactions than with video-mediated interactions, and audio-only interactions seemed more efficient [16,17,18]. Recently, there have been discussions regarding “Zoom fatigue”, a form of exhaustion experienced by participants of video conference meetings [19,20,21,22]. The possible causes of this phenomenon include both a lack of proper social cues (i.e., eye contact, body language), leading to increased cognitive effort, and information overload (i.e., self-image visible, multiple faces visible on the screen), leading to additional stress [23].

A deeper understanding of video-mediated communication can be gained by studying the process of interaction itself [16,24]. Different media provide characteristic constraints and afford specific communicative actions with different degrees of synchronicity. This shapes the ongoing interaction process and, consequently, interaction outcomes. In the case of video-mediated interactions, disrupted social cues and visual information overload may affect the capabilities of nonverbal communication, leading to different coordination dynamics than in face-to-face interactions. We suspect that altered coordination capabilities in online communication may influence informational and affective couplings between participants, may be a possible cause of decreased satisfaction with an interaction, as stated in the recent literature, and may also cause decreased effectiveness of communication as compared to face-to-face interactions.

### 1.1. Dynamics of Video-Mediated Interactions

Patterns of social interaction dynamics are emergent properties shaped by multiple interrelated factors [13,25]. In the case of natural conversation, any change in a participant’s impression of their interlocutor influences the way the participant responds, which in turn influences the interlocutor. This ongoing feedback loop, constituting patterns of interaction dynamics, may work differently in mediated interactions. A communication medium—such as a video-conferencing setup—is one of the factors that may significantly constrain interaction dynamics. In the language of dynamical systems, if a medium offers fewer possibilities for interaction than the number of available options in unmediated communication, the number of degrees of freedom of the system is reduced. On the one hand, when the preferred interaction means are taken away, it may disrupt the interaction. On the other hand, when the redundant modes of communication are reduced, it may present a case of functional reduction in degrees of freedom facilitating the interaction. Either way, the patterns of interaction dynamics are changed.

Constraints imposed by the communication medium can be traced through the analysis of interactions between a person and the medium. In this case, the ecological psychology notion of affordance is helpful [26]. Affordances are opportunities for action and perception offered by the environment to an active subject. They are not simply objective properties of the external objects (shape, size), but meaningful relations in which complementarity between the subject and its environment manifests (graspability, possibility to sit upon). In the social realm, affordances are created and used dynamically by each interactant “on the fly” [27]. Introducing a video-based communication medium creates new possibilities for actions and forms of interaction, while precluding others. The landscape of affordances available for the individuals and the dyad changes, which changes their behavior and cocreated meanings [28].

Affordances of video-mediated interaction are significantly changed by the presence of video latency—a mean delay between the moment the movement is made, and the moment it is visible on another user’s screen. Another aspect is jitter—variability of the delay, caused by the different length of time each data packet takes to arrive. If the jitter is large, movement in the video is not smooth. Video latency during a high quality video call may be 150 ms with 40 ms jitter [29], but these values may vary depending on the network traffic, connection bandwidth and hardware configuration/quality. Since latency works in both directions, the effective time between a communicative action and the perceived response may double. Additionally, glitches in the form of video freezing or distorted images are common during video calls. These factors modify the affordances of interaction participants, for instance, by limiting the possibility of reacting quickly to each other thus constraining their patterns of coordination. It is known that people are able to perceive delays of 200 ms [30,31], which suggests that even relatively small video latency may affect coordination in a video-mediated interaction. Boland et al. [32] studied turn-taking during face-to-face and Zoom conversations and discovered that delays introduced by the latter significantly disrupted the rhythm of conversation, increasing the average turn transition time from 135 ms to 487 ms. Such altered coordination patterns may have further consequences for communication. The length of the gap between turns may provide information on the valence of the upcoming response, with preferred responses coming quicker and taking simpler forms [33]. A gap as short as 300 ms may be sufficient to project that a straightforward acceptance is less probable [34]. Because of the prolonged gaps due to the video latency, speakers may erroneously expect more dispreferred reactions than in face to face communication. Additionally, according to the studies on telephone communication, the longer the delays are, the more interlocutors are perceived as less attentive, less friendly, less extraverted and less conscientious [35].

Another aspect that differentiates video-mediated and face-to-face interactions is the way the image of the conversation partner is presented to the interaction participant. In natural face-to-face conversations, people typically face each other, moving their glances between the face, body and hands of the interaction partner [36], which provides them with specific means to fluently structure the interaction (see, e.g., Rączaszek-Leonardi and Nomikou [37]). In contrast, in a typical video-mediated interaction (for instance, using a laptop computer with a built-in camera), the captured field of vision is much narrower, limiting visual cues concerning whole body movement and hand gestures. This may severely limit nonverbal communication, as hand gestures play an important role in supplementing speech with additional content, disambiguating expressions or organizing turn-taking [38,39,40]. It is possible to compensate for this through the use of other modalities such as head gestures, which are captured well in video-conferencing settings. Head gestures are considered to be important for coordinating interaction, providing confirmatory feedback for the speaker [41] and signaling turn claims [42]. In many cultures head nodding and head shaking are associated with affirmative and negative responses, respectively (Refs. [43,44,45], but with exceptions [46]). Being able to convey approval through head gestures during conversation would be an important factor contributing to the perceived naturalness of an interaction. Additionally, the need to fit within the field of view of the camera may limit the overall movement and induce a feeling of being physically trapped [21]. In face-to-face meetings, people can shift their position and stretch, but during video communication their mobility is limited to a narrow space. This reduced mobility may undermine cognitive performance [47], further disrupting communicative abilities.

Moreover, in many video conferencing programs, there is a setting in which a self-image of the participant is displayed along with the image of their interaction partner. This may be potentially disturbing in several ways. It may change the basic gaze dynamics, which was claimed to serve as a “glue” for interaction [48], and introduce effects on individuals’ behavior similar to the presence of a mirror. Research in social psychology shows that seeing the self-image in a mirror can heighten self-focused attention, which in the case of longer exposition can have negative psychological consequences, including decreased mood or even depression ([19,49,50,51], but see [52,53]). In the interactive context, self-focused attention was reported to decrease prosocial behavior in some contexts [54], although it is possible to find conditions in which it enhances prosocial behavior [55]. Finally, seeing oneself in a mirror provides visual feedback—an additional affordance that might be used for more precise control of one’s appearance and expression. Little is known about the consequences of the visible self-image for coordination with a conversation partner.

Our investigation complements existing studies on the naturalness of online interactions through the introduction of the movement coordination perspective and the dynamical systems methodology, which goes beyond individual cognitive processes by focusing on coupling. We show how the movements of individuals are constrained in video-mediated interactions, and what patterns of interpersonal coordination emerge.

### 1.2. Current Study and Hypotheses

The goal of our study was to explore movement coordination dynamics shaped by the affordances altered by video-mediated means of communication. We identified factors such as: restricted mobility in front of the camera, video latency and jitter, and the optional presence of one’s own mirror image. All these components potentially constrain movement of the individual, modify adopted nonverbal communication strategies, and, finally, reshape interactive patterns of interpersonal coordination. To disentangle the influences of the video medium and the mirror image, we adopted an experimental design in which casual, friendly conversations of the same dyads were recorded in four conditions: (I) video-mediated remote conversation with the mirror image displayed, (II) video-mediated remote conversation without the mirror image, (III) face-to-face conversation with the mirror image, and (IV) face-to-face conversation without the mirror image. We expected the differences to be manifested at the individual level and at the dyadic coordination level. At the individual level:

**Hypothesis** **1.**
*Overall movement will be more restricted in remote interactions, because of the need to stay visible (in the field of view of the camera) and to see the interlocutor.*


**Hypothesis** **2.**
*Intentional communicative gestures will be exaggerated (in comparison to the overall movement) in remote interactions to compensate for potential disruptions.*


**Hypothesis** **3.**
*The availability of the self mirror image in remote interactions will allow participants to calibrate their expressions, making the movement more natural and less exaggerated. No such effect is expected for face-to-face interactions, where natural instantaneous feedback is available through the partner’s reactions.*


Regarding interpersonal movement coordination, we expected that:

**Hypothesis** **4.**
*Coordination will be more stable in face-to-face interactions, and episodes of coordination will be longer.*


**Hypothesis** **5.**
*Coordination will be less stable with the mirror image present, as it presents an additional distraction (participants captivated by their own movement may be less attentive to their partners).*


To operationalize our hypotheses, we tracked participants’ head movement during conversations using OpenPose software [56]. We focused on head movements, as they were important and visible both in face-to-face and remote conversations. According to the existing literature, the dominant head gesture during conversations is nodding, which is associated with vertical motion [40,41]. Head nodding (vertical motion) and head shaking (horizontal motion) are typically distinguished as they are associated with positive and negative responses, respectively [44]. Head nodding was reported to increase the perceived likability and approachability of a person [43]. Following this logic, we decided to differentiate between vertical and horizontal motion in our analyses. After watching the collected video material, we discovered that there were multiple episodes of head nodding in response to the partner, but hardly any head shaking. This was consistent with the friendly character of the conversations, where head nodding is expected to be much more prominent than head shaking [57]. Horizontal head movements in our recordings seemed to result not from head shaking, but mostly from body sways and position adjustments less connected with the conversation dynamics. Thus, at the risk of oversimplification and with the limits of cross-cultural generalization in mind, we interpreted vertical head movement as an indicator of intentional communicative gestures expressing positive reaction to the interlocutor, and horizontal head movement was treated as a control—an indicator of general body movement.

When operationalizing interpersonal coordination, we decided to focus on the congruence of head movement direction within the dyad. Two people moving their heads in the same direction (nodding, tilting, turning, etc.) simultaneously or with a constant delay examplify coordinated behavior. We quantified the coordination using cross-recurrence quantification analysis (cRQA) [58], a nonlinear technique providing measures of coordination stability.

## 2. Materials and Methods

### 2.1. Participants and Setup

The examined material consisted of 24 recordings (137 min in total), collected from interactions of two groups of three people: Group A consisted of three men, and Group B consisted of three women (age 22–35). All participants were university students. The study was approved by the ethics committee of the Faculty of Psychology, University of Warsaw. Participants gave their consent to record their conversations and use them for research purposes.

Participants were students in the same program. Their level of acquaintance was assessed through a short interview. Participants from Group A were attending online courses together and had a chance to get to know each other while doing a group project together. Participants from Group B were engaged in research within the same research group and spent some time socializing before participating in the study. They can be described as colleagues, but there were no close friends within either group. All conversations were held in English, which was the second language for all participants. All participants had previous experience using videoconferencing software and were used to this form of communication.

Within each group, everyone was paired up, therefore creating six dyads (three per group) in total. Each dyad engaged in two conversations: one conducted remotely and one face-to-face, and each of these conversations was divided into two parts: with the mirror image and without. Each part lasted approximately five minutes. We briefed the participants regarding the purpose of the study, length of the conversations and the differences between experimental conditions. Participants knew that their movement will be tracked and their coordination will be analyzed. They were not informed on the detailed study hypotheses. Participants were instructed to keep the conversations casual and choose the topic freely. Most of the conversations started with a general opening question (“What’s up?”) and then developed spontaneously. Topics such as university studies, work, vacations, hobbies, etc., emerged. All conversations were friendly in tone, and no controversial topics or heated debates occurred.

Figure 1 presents the general schema of the four experimental conditions.

The remote conversations took place on the Google Meet platform. Two participants engaged in the conversation, and the researcher joined the meeting and recorded the interaction using OBS Studio software for screen recording. The researcher recorded the meeting in a “gallery view” mode, where images of the two interlocutors were placed side by side. Both participants were recorded with lag characteristic for the videoconferencing platform. In the mirror condition (with self-view), the participants saw both the other person and their own face, while in the no-mirror condition (without self-view) they could only see their interlocutor. They conducted a single 10-min conversation starting without self-view and switching self-view after 5 min. Participants used their own laptops with built-in video cameras.

Before the actual recordings of remote conversations, trial recording sessions took place during which participants were able to familiarize themselves with the setup. After the trial sessions, participants were instructed to adjust their setup (position of the camera, lighting) to improve the quality of the recordings.

Face-to-face conversations were recorded via a smartphone camera connected to a laptop (using Droidcam OBS and OBS studio software). We connected two smartphones to the same laptop via a local WiFi network and used OBS studio to combine the two image streams into a single output video file in which images of two interlocutors were placed side-by-side (as in the typical videoconference setup). We placed each smartphone in front of one of the interlocutors, with the front camera filming one’s face and upper body. The participants were given a few minutes to sit down and adjust their positions to make them feel comfortable and ensure they fit into the video frame. The mirror condition was reproduced by showing the person’s face and upper body position and movements in real time on the smartphone screen. The participants had a single 10-min conversation, in which smartphone screens were dimmed for the first half and were switched on for the second half.

### 2.2. Movement Tracking

We converted the video recordings to a common video format with 20 FPS. Each video frame contained the images of two participants side by side. We cropped the videos to obtain a separate video file for each participant during each conversation. The minimal resolution of the cropped video was 530×304 pixels. All videos were downscaled to this resolution. We processed the videos with OpenPose software [56] to obtain the x-y coordinates of key body parts (see Figure 2). For our analyses we extracted coordinates of points P0 (tip of the nose) and P1 (point in the middle of the torso on the shoulder level). There were missing values due to the algorithm not identifying a keypoint on a particular frame. In the recordings of one male dyad in the remote condition, the numbers of missing values were particularly large (16–55%). We removed these two recordings from the analysis. A small number (<5%) of missing values in other recordings were imputed using linear interpolation. Afterward, we applied a running median filter with a window size of five for each coordinate separately to remove possible outliers.

### 2.3. Measures and Data Analysis Techniques

In our analyses, we focused on the movement of two points: the tip of the nose (point P0), as an indicator of head movement, and the middle of the torso (point P1), as the reference (see Figure 2). To normalize the data, we used the average P0-P1 distance for each person as a natural scale of movement. To operationalize our hypotheses regarding individual movement, we introduced the following measures:Horizontal mobility—standard deviation of the horizontal P0 coordinate divided by the average P0-P1 distance. It is interpreted as a general indicator of participant mobility.Vertical mobility—standard deviation of the vertical P0 coordinate divided by the average P0-P1 distance. It is interpreted as an indicator of communicative nodding gestures.Horizontal-vertical mobility ratio—ratio between horizontal and vertical mobility. It is interpreted as a ratio between overall movement and communicative nodding gestures.

The described measures were calculated separately for each of the two members of the six dyads in each of the four conditions, which should result in 48 data points. Since we excluded two recordings of the particular dyad in the remote condition (see Section 2.2), the final number of analyzed data points was 44.

To analyze the properties of interpersonal coordination, we focused on the direction of frame-to-frame movement of P0 point. For each frame we calculated a 2D vector, representing the shift in position from the previous frame. All vectors were normalized to have unit length. A low-pass Butterworth filter was used to smoothen the data. Then we calculated the interpersonal coordination statistics using the methodology inspired by multidimensional cross-recurrence quantification analysis [59]. We constructed separately for x and y coordinates time-delayed embeddings using a delay of 7 frames and embedding dimension 4 (values chosen using minimal mutual information heuristic for delay and false nearest neighbors for dimension [60]). Embeddings for the two coordinates were concatenated, resulting in a final dataset with eight columns. We constructed a recurrence matrix by calculating distances between all pairs of 8-dimensional vectors and thresholding them using a fixed value. All distances below the threshold formed recurrent points. We chose the threshold value for each matrix separately to ensure that the fraction of recurrence points was always 10%. In this way, RQA statistics were normalized across dyads and experimental conditions (This methodology is different from some other studies using RQA (e.g., Rączaszek-Leonardi et al. [11]), where threshold value is fixed across all samples and the fraction of recurrent points (RR) was compared across conditions. In the case of our data, differences in optimal threshold level were too large for this kind of comparison.).

In layman’s terms, a cross-recurrence matrix represents the temporal structure of “meetings” of two evolving systems. A recurrent point with coordinates (*i*, *j*) means that system *A* at time point *i* was in the same state as system *B* at time point *j*. In the context of participants of our study, recurrence means that two participants moved in the same direction relative to their cameras. Recurrent points on the main matrix diagonal indicate that participants’ movements were synchronized, while recurrent points outside the main diagonal indicate more complex kinds of coordination. We controlled for the fraction of recurrent points—denoting the overall strength of coordination— and quantified characteristic patterns of coordination through the analysis of diagonal and vertical lines formed by recurrent points. We will use the following notation: *l*—length of diagonal line, P(l)—probability of a diagonal line of length *l* occurring, *v*—length of vertical line, P(v)—probability of a vertical line of length *v* occurring. Then, popular recurrence quantification measures can be defined as follows:Determinism, fraction of recurrent points forming diagonal lines.
$$DET=\frac{\sum_{l=l_{min}}^{N}lP\left(l\right)}{\sum_{l=1}^{N}lP\left(l\right)}$$A large DET means that there are stable episodes of coordination and that coordination is more predictable. In interaction it suggests that partners may anticipate each other’s actions and successfully maintain coordination.Entropy of the distribution of diagonal line lengths.
$$ENTR=-\sum_{l=l_{min}}^{N}P\left(l\right)\ln P(l)$$A large ENTR means that the coordination is more complex with more characteristic patterns of coordination. This suggests that the interaction process is more varied.Average length of a diagonal line.
$$L=\frac{\sum_{l=l_{min}}^{N}lP\left(l\right)}{\sum_{l=l_{min}}^{N}P\left(l\right)}$$A large L means that the episodes of coordination are longer on average.Lmax – maximum length of a diagonal line. A large Lmax means that it is possible to maintain coordination for a longer time.Laminarity, fraction of recurrent points forming vertical lines.
$$LAM=\frac{\sum_{v=v_{min}}^{N}vP\left(v\right)}{\sum_{v=1}^{N}vP\left(v\right)}$$Vertical lines form when one participant remains in the same state (moving uniformly or being still) for some time. A large LAM indicates that participants’ movement is steadier.Trapping time, average length of a vertical line.
$$TT=\frac{\sum_{v=v_{min}}^{N}vP\left(v\right)}{\sum_{v=v_{min}}^{N}P\left(v\right)}$$A large TT means that the episodes of steady movement are longer on average.

We counted only diagonal and vertical lines of length 10 or more ($l_{min}=v_{min}=10$ corresponds to episodes of coordination or steady movement lasting 0.5 s or more; this value was chosen empirically to ensure sufficient variability of DET and LAM statistics). RQA measures were calculated for each of the 6 dyads across 4 conditions, except for the one dyad for which recordings of remote interactions were excluded from the analysis (see Section 2.2). The final sample consisted of 22 observations.

We performed statistical analysis using mixed-effects linear models adequate for the repeated measures experimental design. All analyses were performed in Julia programming language using the packages DynamicalSystems.jl [61] and MixedModels.jl [62].

## 3. Results

### 3.1. Horizontal and Vertical Mobility

We started by comparing participants’ mobility along horizontal and vertical dimensions across the experimental conditions (see Figure 3). The differences were quantified using mixed-effects linear models, with model coefficients presented in Table 1.

Horizontal mobility was similar in all conditions, vertical mobility was larger in remote conditions (p=0.001), and the ratio was significantly larger in face-to-face conditions (p<0.001). Additionally, studying the plot (Figure 3b) suggested that vertical mobility might be slightly larger in the “remote no-mirror” condition in comparison to the “remote mirror” condition. To verify this, we applied an additional paired samples Student’s *t*-test which compared the two conditions. We obtained t=−4.8922 (DF=10) and p<0.001, which gives support to the hypothesis that the conditions differ.

Interpreting the results in the light of research hypotheses, we had to reject Hypothesis 1, as neither horizontal nor vertical mobility was visibly restricted in remote interactions. Hypothesis 2—stating that in remote interaction, participants exaggerate communicative gestures—was confirmed by the differences in vertical mobility and horizontal-vertical mobility ratio. Larger vertical mobility and a smaller ratio in remote conditions suggest that participants increased their range of nodding movements while restricting other movements. Finally, comparison of vertical mobility between the “remote mirror” and “remote no-mirror” conditions supports Hypothesis 3: the presence of self-image in the mirror condition reduced exaggerated nodding gestures.

### 3.2. Interpersonal Movement Coordination

Figure 4 presents cRQA statistics for interactions in all four conditions, while Table 2 contains coefficients of mixed-effects linear regression models verifying the strengths of effects for each statistic. As we can see, differences between remote and face-to-face interactions are evident on all measures except TT, which is congruent with Hypothesis 4. We found no visible effect of mirror image presence on movement coordination; there is no support for Hypothesis 5.

## 4. Discussion

Our results show that shifting to remote communication changes the dynamics of movement manifested by individuals and on the dyadic level. Our intuitions that participants move differently during remote and live interactions were confirmed by quantitative analyses. During remote interactions, they exaggerated their nodding gestures, which may stem both from the awareness of their lesser visibility by the partner and compensation for the unnaturalness of the situation. This effect was reduced when the self-image was present. One of the possible explanations is that in remote conversations participants lacked some immediate feedback from their interlocutors and were unsure whether their gestures were visible. The self-image might have provided compensatory feedback allowing them to calibrate their expression.

On the dyadic level, we demonstrated that in video-mediated remote conversations interpersonal coordination was less stable (smaller DET), less complex (smaller ENTR) and occurred in shorter episodes (smaller L and Lmax). Our findings suggest that partners interacting remotely do not form a coupled system with the same properties as in natural face-to-face interactions. According to De Jaegher et al. [13], particular dynamics of social interaction enable processes of social cognition. With the altered interaction dynamics, these processes might be disrupted, diminishing mutual understanding between interaction partners.

Contrary to our expectations, the presence of the self-image in the mirror condition had no visible effect on movement coordination. It is possible that the movements we captured do not reflect the changes that might be induced by this presence—such as changes in gaze behavior. In any case, these changes did not result in altered coordination. It is also possible that the effects were too subtle to be detected in the current experimental design, e.g., due to the brevity of five-minute conversations.

The study of movement coordination not only provided objectively measurable determinants of the quality of communication but also allowed us to transfer the analysis from the level of the individual to the level of dyad dynamics. This is in line with the embodied and enacted perspectives on social interactions [2,13] and compatible with Burgoon’s “principle of interactivity” [16], suggesting that the process of interaction afforded by a communication medium should be characterized first before investigating interaction outcomes. Our investigation of movement coordination complements individualistic studies pertaining to individual satisfaction and cognitive load during online conversations [19,20].

The interactive perspective might potentially provide an alternative explanation of the “Zoom fatigue” phenomenon. Our results demonstrate that interaction properties deemed to enable social cognition [13] are altered, and the coordination is overall less complex (smaller DET and ENTR) in remote interactions. In that case, what is missing are not so much individual social cues (such as gestures or facial expressions) but rather “interactive cues”—specific properties of the interaction dynamics that allow us to tell an affiliative conversation from a quarrel, the continuation of an ongoing conversation topic from the beginning of a new topic, etc. Lack of this interactional scaffolding might lead to confusion and frustration. Further research could test this hypothesis by combining the two perspectives and checking how the satisfaction reported by the respondents participating in video-mediated interactions is reflected in their coordination. This would confirm whether coordination properties are actually connected with the experienced fatigue. The results could also be compared with previous studies associating movement synchrony with positive outcomes in face-to-face interactions [7,8,63].

Another intriguing perspective that may provide a framework for reflection on the sources of perturbations in video communication is the comparison with audio-only communication. From an information theory perspective, a video call offers a channel of greater capacity—it allows transmitting more information than a phone call. However, despite audio communication being more limited, we observe no “phone fatigue” phenomenon. This may suggest that perhaps extra information provided by video communication is actually more cognitively demanding than helpful. For the “receiver”, the nonverbal message might be more difficult to interpret because some contextual social cues facilitating the interpretation are altered in remote interactions (for instance, response times allowing the prediction of positive or negative reactions [34]). Increased channel capacity in the case of video-mediated interactions may also be more demanding from the sender’s perspective. For example, being aware that at least some parts of their body are visible to the partners and therefore gestures are an important source of information on interaction, senders feel obliged to use their body language in the same manner as in a normal face-to-face conversation. This makes a difference with audio-only interactions, since the same body language that is appropriate during a phone call is no longer appropriate within video conversation. At the same time, remote communication limits the possibility of the natural use of body language, as demonstrated by our result of more exaggerated nodding gestures, which may lead to an experience of frustration or fatigue. Examination of the impact of these factors in comparison between video and audio-only conversations is another interesting line of further research, especially with an attempt to untangle the experience related to sender and receiver perspectives.

Continuing the information-theoretic considerations, we should also discuss the role of noise in the communication channel or the reliability of a medium. From the user point of view, a tool that offers less functionality but is more predictable is still more effective than a more powerful but unreliable tool [64]. A video call is a channel of greater capacity than an audio-only call, but at the same time, it is more affected by noise due to latency and jitter. Video calls are prone to image and audio lags and disturbances, which even if they are minor and seemingly insignificant, may keep both sender and receiver in a state of constant uncertainty about how much information is lost during the transmission. Shorter episodes of stable coordination in video-mediated interaction discovered in the study may be a sign of low reliability of this medium: whenever participants began to coordinate on a nonverbal level, an unpredictable signal distortion might have destroyed the coordination.

Our small exploratory study does not allow us to formulate any strong recommendations concerning preferred forms of remote communication. Nevertheless, some cautious observations can be formulated. Despite worries that the presence of the self-image makes the conversation less natural, it may have its use as a source of compensatory feedback during interaction. Using this option can thus be recommended. As coordination in remote interaction is overall less stable, some conscious effort could be made to stabilize it. The simplest idea would be to deliberately slow down and avoid fast gestures, which could be misinterpreted due to video lag. Assessing such a strategy would require additional studies.

### Limitations

Although our results confirmed that movement coordination is impaired in remote communication, they do not allow us to draw conclusions as to the main factors that contribute to this result. We note that our data are not conclusive on the effect of the presence of the self-image in the mirror condition, which might be one possible source of distraction. We observed no significant differences in movement coordination comparing these two conditions of both live and remote conversations; the ineffectiveness of the variable manipulation may be the underlying reason. The participants, being aware that they were being recorded, might have a lower tendency to focus attention on their image than in a natural environment. Additionally, looking in a mirror while talking to someone across the table is much less natural than seeing one’s own image during a video call, which could have had an impact on our results in face-to-face conversations. Furthermore, as our sample was very small, it did not allow us to study interindividual differences in responses to online interactions.

The study can be extended through tracking whole body position during conversations and including hand gestures, body positions, etc., in the analysis. It would be possible to supplement coordination measures with the measure of behavior matching, that is body position mirroring [9]. Specific gestures or expressions could be identified automatically using machine learning techniques [65]. To better render the differences in coordination in remote and live interactions it would be crucial to obtain measures on other “coupling means” in dyadic conversations than the movement itself, such as gaze coordination and vocal dynamics. Related to body movement coordination they would inform about the use of the relevant cues as affordances for interaction and allow for forming a fuller picture of the relevant differences.

## 5. Conclusions

The differences between video-mediated and face-to-face interactions cannot be explained by either the technical properties of the medium or individual cognitive processes alone. In this study, we tried to apply an interactive perspective to identify key factors shaping our experience of online interactions. In line with this perspective, our study revealed significant differences in patterns of interlocutors’ coordination between video-mediated remote and live interactions. We demonstrated that in video communication, the stability of movement coordination is lower, which may have a negative impact on the overall quality of interaction. The presence of the mirror image did not have a detectable effect on coordination; however, it seems that the mirror image helped to control one’s expression during remote interactions, making the communicative gestures less exaggerated. Vast differences in coordination patterns indicate that the remote medium radically alters the landscape of affordances for communicative actions. It remains to be seen which affordances result in those differences when they are altered.

## Figures and Tables

**Figure 1 entropy-24-00559-f001:**
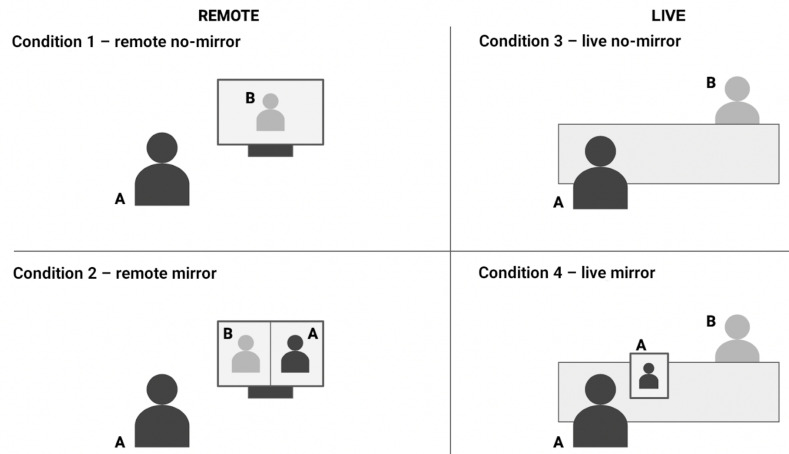
General schema of experimental conditions. In Condition (1), “remote no-mirror”, the participant sees their partner on the screen; in Condition (2), “remote mirror”, the participant sees their partner and their own mirror image side by side. In Condition (3), the “live no-mirror” participant sits in front of their partner with a dimmed smartphone screen placed in between, and in Condition (4), the “live mirror” participant sits in front of their partner with a smartphone displaying mirror image placed in between.

**Figure 2 entropy-24-00559-f002:**
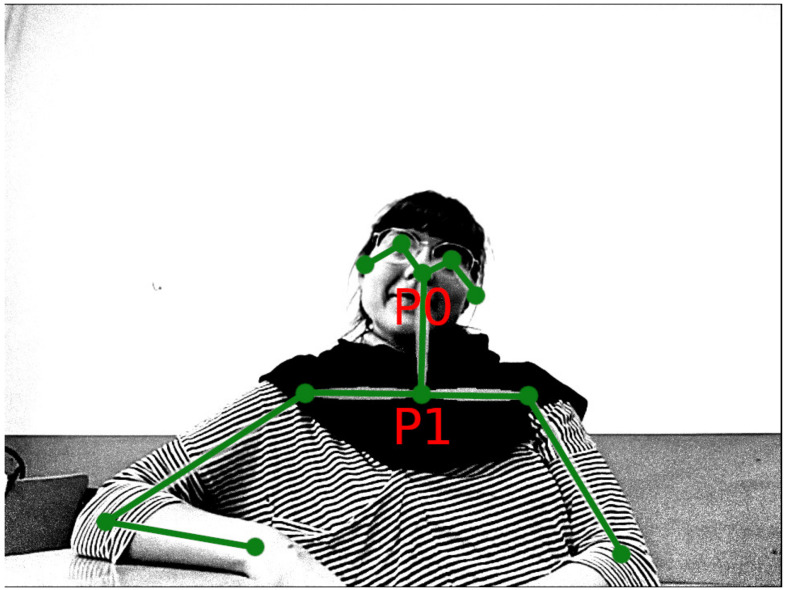
Output from OpenPose program: a video frame with detected key points marked. The two key points used in our analysis are P0 (tip of the nose) and P1 (point in the middle of the torso on the shoulder level).

**Figure 3 entropy-24-00559-f003:**
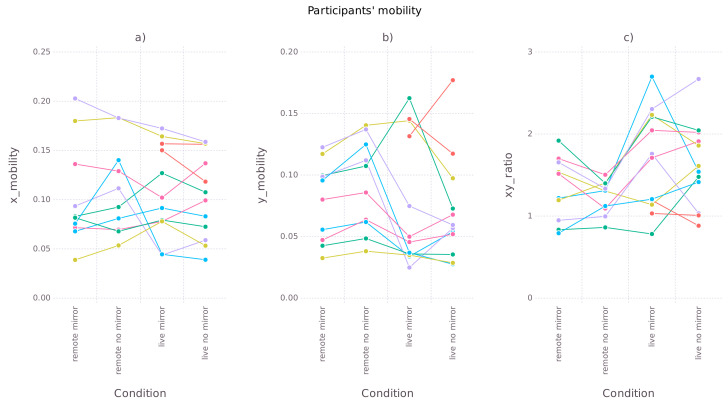
Average participants’ mobility in horizontal (**a**) and vertical (**b**) dimensions, and their ratio (**c**) across experimental conditions. Mobility is defined as the standard deviation of the participant position on the video frame. For each dyad, two lines are drawn: one for Participant A, and one for Participant B (same color lines for participants in each dyad).

**Figure 4 entropy-24-00559-f004:**
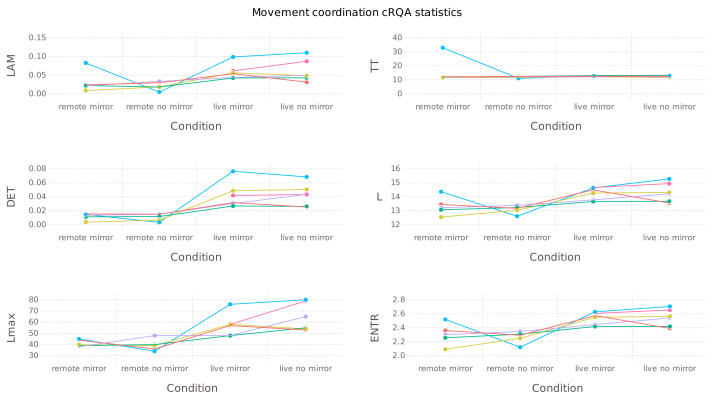
cRQA statistics describing properties of participants’ movement coordination across experimental conditions. For each dyad a single line is drawn.

**Table 1 entropy-24-00559-t001:** Coefficients of mixed-effects linear models comparing horizontal and vertical mobility across experimental conditions.

	Est.	SE	z	*p*	*σ*
Horizontal mobility					
(Intercept)	0.1046	0.0126	8.27	<10^−15^	0.0398
remote	0.0088	0.0066	1.33	0.1825	
no mirror	0.0015	0.0063	0.24	0.8104	
Residual	0.0210				
Vertical mobility					
(Intercept)	0.0724	0.0123	5.87	<10^−8^	0.0374
remote	0.0235	0.0074	3.19	0.0014	
no mirror	0.0024	0.0071	0.35	0.7295	
Residual	0.0235				
Horizontal-vertical mobility ratio					
(Intercept)	1.6989	0.1281	13.27	<10^−39^	0.3627
remote	−0.4709	0.0910	−5.18	<10^−6^	
no mirror	−0.0827	0.0876	−0.94	0.3450	
Residual	0.2906				

**Table 2 entropy-24-00559-t002:** Coefficients of mixed-effects linear models comparing various RQA measures across experimental conditions.

	Est.	SE	z	*p*	*σ*
ENTR					
(Intercept)	2.5486	0.0362	70.41	<10^−99^	0.0000
remote	−0.2672	0.0418	−6.39	<10^−9^	
no mirror	−0.0138	0.0418	−0.33	0.7408	
Residual	0.1024				
DET					
(Intercept)	0.0429	0.0042	10.23	<10^−23^	0.0043
remote	−0.0323	0.0044	−7.33	<10^−12^	
no mirror	−0.0001	0.0044	−0.03	0.9776	
Residual	0.0108				
L					
(Intercept)	14.3208	0.1730	82.78	<10^−99^	0.1239
remote	−1.1179	0.1910	−5.85	<10^−8^	
no mirror	−0.0553	0.1910	−0.29	0.7720	
Residual	0.4679				
Lmax					
(Intercept)	59.5000	2.9122	20.43	<10^−92^	1.5260
remote	−21.5000	3.2848	−6.55	<10^−10^	
no mirror	2.8333	3.2848	0.86	0.3884	
Residual	8.0462				
LAM					
(Intercept)	0.0593	0.0088	6.71	<10^−10^	0.0120
remote	−0.0315	0.0085	−3.71	0.0002	
no mirror	0.0024	0.0085	0.28	0.7764	
Residual	0.0208				
TT					
(Intercept)	13.0525	1.5470	8.44	<10^−16^	0.0000
remote	2.1017	1.7863	1.18	0.2394	
no mirror	−0.9629	1.7863	−0.54	0.5899	
Residual	4.3755				

## Data Availability

The data presented in this study are openly available in OSF at DOI 10.17605/OSF.IO/8YA47.

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
