# Peer review of "Dynamics of Remote Communication: Movement Coordination in Video-Mediated and Face-to-Face Conversations"

_entropy, 2022, doi:10.3390/e24040559_

Round 1
Reviewer 1 Report
This is an interesting study on a relevant topic of clear importance in the current context of massively video-mediated social interaction. The authors builds upon a solid methodology and the paper is generally ell argued and well written. However, the design of the experiment has some major weaknesses in the preparatory phase that need to be carefully clarified for the paper to be publishable. In the revised version, the authors should carefully and convincingly address the following points.
The identification of vertical movement as an indicator of intentional communicative gestures and of horizontal movements as an indicator of spontaneous movement could make sense in principle (but would be clearly problematic an inter-cultural setting, for instance), but needs to be validated in some way or at least must find support in the existing literature. As this is a central assumption of the study, it cannot be simply posited without a strong argument to justify it. Even if the subjects in the present study are culturally homogeneous, we cannot simply assume such a major point as self-evident.
The dyads are characterized by acquaintances of varying degree. One should control for degree of acquaintance, because this clearly influences the nature, tone and content of communication and reflects into bodily signals. If degree of acquaintance is measurable to some extent, this should be accounted for in the analysis. If it is not, this may be a major problem in an analysis that focuses on the micro-characteristics of the dyadic postural interaction.
What kind of briefing the subjects received before the conversation? Was there a protocol? Could they choose freely which topics to touch upon? This is again an essential feature for the sake of the subsequent interaction and should be specified much more carefully.
Recording clearly affects the naturalistic condition of the experiment. The experiment design did not consider ways to mitigate this effect (e.g. by making people accustomed to recorded interaction through preliminary sessions, etc.). The awareness of being recorded during a conversation that is not intrinsically motivated but induced can greatly influence posture, gestures, etc. The authors should address this point carefully, possibly making reference to relevant literature to better defend their approach on this point.
Although the results are objectively of interest, how they may help us better understand the phenomenon of Zoom fatigue (provided that it exists) is not clear from the text. The authors should provide a clearer and sharper argument here.
Author Response
Rev1: The identification of vertical movement as an indicator of intentional communicative gestures and of horizontal movements as an indicator of spontaneous movement could make sense in principle (but would be clearly problematic an inter-cultural setting, for instance), but needs to be validated in some way or at least must find support in the existing literature.
Re: Thank you for making this important point. The operationalization that we decided upon is definitely a simplification, which seemed to make sense in the light of our material but should not be accepted as universal. However, clear distinction between vertical and horizontal head movement is made by multiple studies, and the privileged role of vertical head movements (nodding) as a general positive reaction, acceptance and confirmation during backchanneling. We included adequate references. We hope that it makes our case stronger.
In the revised introduction:
Head gestures are considered to be important for coordinating interaction, providing confirmatory feedback for the speaker (Wlodarczak et al. 2012) and signaling turn claims (Duncan 1972). In many cultures head nodding and head shaking are associated with affirmative and negative responses, respectively (Osugi and Kawahara (2018); Moretti and Greco (2018, 2020), but with exceptions, (Andonova and Taylor 2012)). Being able to convey approval through head gestures during conversation would be an important factor contributing to the perceived naturalness of an interaction.
In the revised study description:
To operationalize our hypotheses, we tracked participants’ head movement during conversations using OpenPose software (Cao et al. 2021). We focused on head movements, as they were important and visible both in face-to-face and remote conversations. According to the existing literature, the dominant head gesture during conversations is nodding, which is associated with vertical motion (Wagner, Malisz, and Kopp 2014; Wlodarczak et al. 2012). Head nodding (vertical motion) and head shaking (horizontal motion) are typically distinguished as they are associated with positive and negative responses, respectively (Moretti and Greco 2018). Head nodding was reported to increase the perceived likability and approachability of a person (Osugi and Kawahara 2018). Following this logic, we decided to differentiate between vertical and horizontal motion in our analyses. After watching the collected video material, we discovered that there were multiple episodes of head nodding in response to the partner, but hardly any head shaking. This was consistent with the friendly character of the conversations, where head nodding is expected to be much more prominent than head shaking (Fusaro, Vallotton, and Harris 2014). Horizontal head movements in our recordings seemed to result not from head shaking but mostly from body sways and position adjustments less connected with the conversation dynamics. Thus, at the risk of oversimplification and with the limits of cross-cultural generalization in mind, we interpreted vertical head movement as an indicator of intentional communicative gestures expressing positive reaction to the interlocutor, and horizontal head movement was treated as a control—an indicator of general body movement.
_______
Rev1: The dyads are characterized by acquaintances of varying degree. One should control for degree of acquaintance, because this clearly influences the nature, tone and content of communication and reflects into bodily signals.
Re: Thank you for that remark. The phrasing that we used in the original manuscript was unfortunate. We verified the degree of acquaintance through short interviews when recruiting participants. We elaborate on that in the revised manuscript:
Participants were students in the same program. Their level of acquaintance was assessed through a short interview. Participants from Group A were attending online courses together and had a chance to get to know each other while doing a group project together. Participants from Group B were engaged in research within the same research group and spent some time socializing before participating in the study. They can be described as colleagues, but there were no close friends within either group. All conversations were held in English, which was the second language for all participants. All participants had previous experience using videoconferencing software and were used to this form of communication.
We decided that relations in both groups are qualitatively similar and thus are comparable in the context of our small study. We agree that extending the study group would call for some other method for controlling the degree of acquaintance.
Rev1: What kind of briefing the subjects received before the conversation? Was there a protocol? Could they choose freely which topics to touch upon?
Re: Our idea was to keep the conversations as natural as possible. Since in the trial sessions conversations tended to start naturally, we left the choice of the conversation topic to the participants. In the revised manuscript we describe the briefing given to participants in more detail:
We briefed the participants regarding the purpose of the study, length of the conversations and the differences between experimental conditions. Participants knew that their movement will be tracked and their coordination will be analyzed. They were not informed on the detailed study hypotheses. Participants were instructed to keep the conversations casual and choose the topic freely. Most of the conversations started with a general opening question (“What’s up?”) and then developed spontaneously. Topics such as university studies, work, vacations, hobbies, etc., emerged. All conversations were friendly in tone, and no controversial topics or heated debates occurred.
____
Rev1: Recording clearly affects the naturalistic condition of the experiment. The experiment design did not consider ways to mitigate this effect (e.g. by making people accustomed to recorded interaction through preliminary sessions, etc.).
Re: We understand that concern. While we failed to mention this in the original manuscript, we indeed ran test recording sessions with all the participants to make them accustomed to the procedure. Additionally, all our participants had previous experience with remote interactions, as they had already had experienced over a year of online meetings with cameras on, either for work or classes, many of which were streamed or recorded. Please see the description in the revised manuscript:
All participants had previous experience using videoconferencing software and were used to this form of communication.
[…]
Before the actual recordings of remote conversations, trial recording sessions took place during which participants were able to familiarize themselves with the setup. After the trial sessions, participants were instructed to adjust their setup (position of the camera, lighting) to improve the quality of the recordings.
While we did not conduct a formal post-experiment interview or structured questionnaire, our participants reported that they had had quickly got used to the camera/the recording phone, and focused mostly on the interlocutor. We believe this should ease some of the concern regarding the recording.
_______________________
Rev1: Although the results are objectively of interest, how they may help us better understand the phenomenon of Zoom fatigue (provided that it exists) is not clear from the text. The authors should provide a clearer and sharper argument here.
Re: We mention the Zoom fatigue phenomenon to position our study in the current research and stress the importance of the studies of naturalistic remote conversation. Our study cannot directly contribute to the knowledge on Zoom fatigue since we did not measure satisfaction/frustration of our participants. Still, we believe that the interactive perspective in general might provide a useful framework for studying this phenomenon. We elaborate on that point in the revised manuscript:
The interactive perspective might potentially provide an alternative explanation of the “Zoom fatigue” phenomenon. Our results demonstrate that interaction properties deemed to enable social cognition (De Jaegher, Di Paolo, and Gallagher 2010) are altered, and the coordination is overall less complex (smaller DET and ENTR) in remote interactions. In that case, what is missing are not so much individual social cues (such as gestures or facial expressions) but rather “interactive cues” – specific properties of the interaction dynamics that allow us to tell an affiliative conversation from a quarrel, the continuation of an ongoing conversation topic from the beginning of a new topic, etc. Lack of this interactional scaffolding might lead to confusion and frustration. Further research could test this hypothesis by combining the two perspectives and checking how the satisfaction reported by the respondents participating in video-mediated interactions is reflected in their coordination. This would confirm whether coordination properties are actually connected with the experienced fatigue. The results could also be compared with previous studies associating movement synchrony with positive outcomes in face-to-face interactions (Bernieri 1988; Latif et al. 2014; Tschacher, Rees, and Ramseyer 2014).
Reviewer 2 Report
The paper describes an experiment focused on testing differences in moviment coordination durign peer conversations in four different scenarios: video-mediated and face-to-face conversations and mirrored and non-mirrod conversations.
The authors placed the importance of their study under the current zoom fatigue situation, where different studies have being demonstrated the pandemic period has fostered a number difficulties and limitations faced by users during remote communication.
For testing their hypothesis about the differences of movement in the distinct scenarios the authors recorded the conversations and then preprocessed them with OpenPose software. The measures for the analysis considered the horizontal and vertical mobility and their ratio. For the analysis they build a cross-recurrent matrix to represent the temporal structure of the two systems (meetings) and calculated five measures (recurrence quantification measures): DET, ENTR, L, Lmax, and LAM.
After the analysis the authors observed that the dynamics of movements between remote and F2F conversations are different, and that in remote conversations the participants exaggerate their nodding gestures to compensate the lack of the natural feedback they normally have during F2F conversations.
The paper is well written, it tackles an important issue of the current days and it sounds methodologically fine. I would suggest the authors to incorporate more possibilities of future work, such as considering corporal postures as a whole using machine learning techniques for this identification and different methodologies for the comparison among the 4 distinct scenarios. A lot has being developed, for instance, in the field of Multimodal Learning Analytics that could serve as interesting inspiration for the authors to continue their work. Moreover, the work lacks some sort of recommendation regarding which scenario people should prefer/adopt. I am not sure if this is possible at the current state of the experimentation, but it would certainly enrich the paper to hear a bit more about the adaptations people are experimenting in order to lesser the remote fatigue. These recommendations could related to the settings of the video-conferencing tools, or even venturing possible ideas for new functionalities.
At last, I would recommend the authors to provide more in depth information about the technological setup required to run the experiments as well as a more in depth explanation about the recurrence quantification measures, their meaning and calculation.
Author Response
Rev 2: I would suggest the authors to incorporate more possibilities of future work, such as considering corporal postures as a whole using machine learning techniques for this identification and different methodologies for the comparison among the 4 distinct scenarios.
Re: We are grateful for this suggestion. We see such possibilities, indeed we already discussed a possibility to include multimodal data involving eye gaze and vocalizations. Of course, including whole body posture in the analysis is a natural extension of our work. We mentioned that in the revised discussion:
The study can be extended through tracking whole body position during conversations and including hand gestures, body positions, etc., in the analysis. It would be possible to supplement coordination measures with the measure of behavior matching, that is, body position mirroring (Fujiwara and Daibo 2021). Specific gestures or expressions could be identified automatically using machine learning techniques (Beugher, Brône, and Goedemé 2018).
Rev 2: Moreover, the work lacks some sort of recommendation regarding which scenario people should prefer/adopt.
Re: As this is initial exploratory research using a very small group of subject, we are very cautious about formulating recommendations. Encouraged by the reviewer comment, we included a short paragraph in the discussion:
Our small exploratory study does not allow us to formulate any strong recommendations concerning preferred forms of remote communication. Nevertheless, some cautious observations can be formulated. Despite worries that the presence of the self-image makes the conversation less natural, it may have its usage as a source of compensatory feedback during interaction. Using this option can be thus recommended. As coordination in remote interaction is overall less stable, some conscious effort can be made to stabilize it. The simplest idea would be to deliberately slow down and avoid fast gestures, which could be misinterpreted due to video lag. Assessing the viability of such a strategy would require additional studies.
Rev 2: At last, I would recommend the authors to provide more in depth information about the technological setup required to run the experiments as well as a more in depth explanation about the recurrence quantification measures, their meaning and calculation.
Thank you for this recommendation. Following it, we incorporated actual mathematical formulas behind RQA measures and elaborated a little bit on their meaning. Also, we extended the setup description slightly.
Round 2
Reviewer 1 Report
The authors did an excellent job in revising the paper and I congratulate them for this. The paper is now publishable. I only recommend a final reading to eliminate a few remaining stylistic bugs (e.g. line 254: "During live face-to-face conversations, the conversations...").